# Multilingual Pixel Representations for Translation and Effective Cross-lingual Transfer

**Elizabeth Salesky**[J]   **Neha Verma**[J]   **Philipp Koehn**[J]   **Matt Post**[H, M]

[J]Johns Hopkins University
[H]Human Language Technology Center of Excellence
[M]Microsoft

esalesky@jhu.edu

## Abstract

We introduce and demonstrate how to effectively train multilingual machine translation models with pixel representations. We experiment with two different data settings with a variety of language and script coverage, demonstrating improved performance compared to subword embeddings. We explore various properties of pixel representations such as parameter sharing within and across scripts to better understand where they lead to positive transfer. We observe that these properties not only enable seamless cross-lingual transfer to unseen scripts, but make pixel representations more data-efficient than alternatives such as vocabulary expansion. We hope this work contributes to more extensible multilingual models for all languages and scripts.

## 1 Introduction

Multilingual model vocabularies are finite and typically smaller than the possible set of Unicode characters, inherently leaving some languages and scripts under-represented. As coverage increases, parameter allocation to each language decreases, resulting in a trade-off between capability, capacity, and coverage. Recent work on pixel representations (Salesky et al., 2021; Rust et al., 2023) provides an appealing alternative to past approaches, because they do not have a discrete model vocabulary or finite embedding matrix, and can represent all scripts with complete parameter sharing.

Recent work (Rust et al., 2023) has also shown that pixel-based models can be directly finetuned across scripts without vocabulary extensions, adapters, or transliteration. However, pixel representations have previously only been trained or finetuned on individual languages at a time, rather than multilingually. This leaves unanswered questions about the effects of multilingual co-training, such as whether similar scripts will interfere with or boost performance, or if architectural changes will be needed given the larger input space.

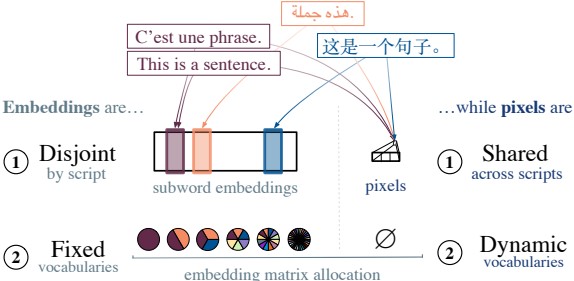

Figure 1: Embedding matrices are disjoint parameter allocations by script, leading to a vocabulary bottleneck. Pixel representations however share parameters across scripts and are not limited to a discrete vocabulary.

In this work we demonstrate how to effectively parameterize and train multilingual translation models with pixel representations, leading to improvements of up to 9 BLEU on two multilingual datasets with diverse language and script coverage. We explore various properties of pixel representations in order to understand their potential benefits and limitations, including positive transfer and representational similarity between languages, parameter sharing, and frequency-based relationships. Finally, we show that not only can pixel representations be finetuned cross-lingually or to unseen scripts, but can do so more data-efficiently than alternatives such as vocabulary expansion, with significant improvements for unseen scripts.

## 2 Our approach

Covering the larger character sets[1] in multilingual models commonly results in significant parameter increases in the embedding matrix and softmax, creating a *vocabulary bottleneck*. While sampling data by language to balance vocabularies is common for large-scale multilingual systems (Fan et al., 2021), sampling may cause common vocabulary to be out-of-vocabulary (OOV) for languages with longer-tail character distributions like Chinese (NLLB Team

---

[1]There are 143,698 Unicode character codepoints as of v13.0.

et al., 2022).[2] One alternative is to move to byte-based representations, which combats exploding model parameters by reducing the set of embeddings to 256. However, this approach increases sequence lengths up to $12\times$ compared to characters, determined by the script's Unicode encoding, making optimal batch sizes prohibitively large and slow for our computational resources.

Rendering text to images bypasses many of the vocabulary challenges posed by multilingual modeling. Pixel-based representations have the advantage of no predetermined static vocabularies, no exploding embedding matrix parameters or sequence lengths, and complete parameter sharing across similar word forms at a sub-character level regardless of the underlying Unicode or byte structure.

Below we present the technical details of our approach and comparisons before proceeding to experimental settings and results.

## 2.1 Encoding text with pixels

Figure 2 demonstrates the rendering process and resulting Transformer inputs. We render text using the PangoCairo library[3,4] following Rust et al. (2023) with a font size of 10pt at 120 DPI. We tokenize sentence-level images into fixed-size image tokens with $h=24$, $w=24$, and stride $s=12$, which results in ~3 Latin characters per token. The height was chosen to fit the wide variety of scripts and diacritics in our experimental data with a fixed font size. We use the Google Noto Sans fonts collection which covers the majority of Unicode codepoints.[5] Further discussion on rendering parameter choices is found in App. C. No preprocessing is applied before rendering. We train many-to-one multilingual models with pixel representations on the source side, and generate discrete subword tokens as the target as below.

## 2.2 Traditional subword tokenization

We generated all subword vocabularies using SentencePiece unigramLM (Kudo, 2018; Kudo and Richardson, 2018). In exploratory experiments, we

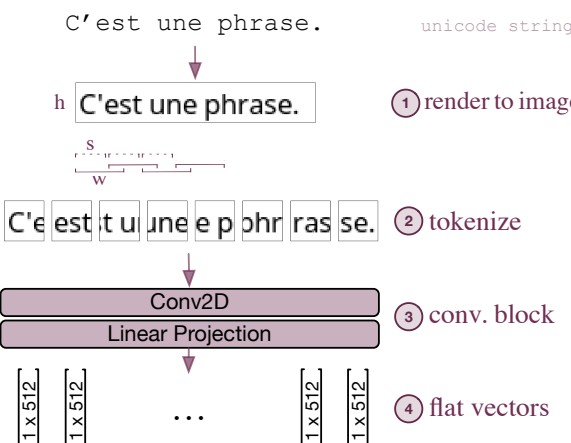

Figure 2: **Encoding text with pixels:** text is rendered to images by sentence. Image tokens are created by overlapping sliding windows of fixed height ($h$), width ($w$), and stride ($s$). Convolutional layer output is projected to flat vectors for subsequent Transformer layers.

compared the union of subword vocabularies constructed per-language to a jointly-trained subword vocabulary of the same total size. Individual vocabularies were of size 5k,[6] and scaled equivalently for joint vocabularies, e.g. 35k for 7 source languages. The two constructions did not result in significant differences in downstream performances in our balanced or imbalanced data settings so we present only joint vocabulary results in the main text, as this approach scales more easily to 59 languages. Results for both constructions are shown in App. G. We use separate source and target vocabularies and share target vocabularies between subword and pixel models in order to isolate the source representation change. Vocabulary sizes for all models and datasets are shown in Table 4 in App. B.

## 2.3 Model architecture

Our core architecture follows Salesky et al. (2021) and combines a convolutional block[7] which processes image tokens and produces flattened vectors (Figure 2) with a Transformer encoder-decoder model. Convolutional layers use one color channel and a $3 \times 3$ kernel with a stride of 1. Our conventional text models share the same Transformer architecture and replace the convolutional block with a traditional embedding matrix of size $V \times 512$.

Our base models are Transformers with 6 encoder and 6 decoder layers each, with hidden units of dim 512, feed-forward layers of dim 1024, and

---

[2]For example, the NLLB model vocabulary does not include the common characters in 'mother' in Chinese, 妈妈.

[3]https://docs.gtk.org/PangoCairo

[4]PangoCairo provides greater flexibility than alternatives such as PyGame, used in previous work, by supporting fallback fonts at the character level. This is necessary not only for code-mixing but to support common occurrences such as non-transliterated entities within non-Latin scripts.

[5]See https://notofonts.github.io/overview for the Noto fonts and their Unicode coverage.

---

[6]Based on tuning for the same dataset in Salesky et al. (2021).

[7]A 2D convolutional layer followed by 2D batch normalization, a ReLU layer, and a linear projection to $1 \times 512$.

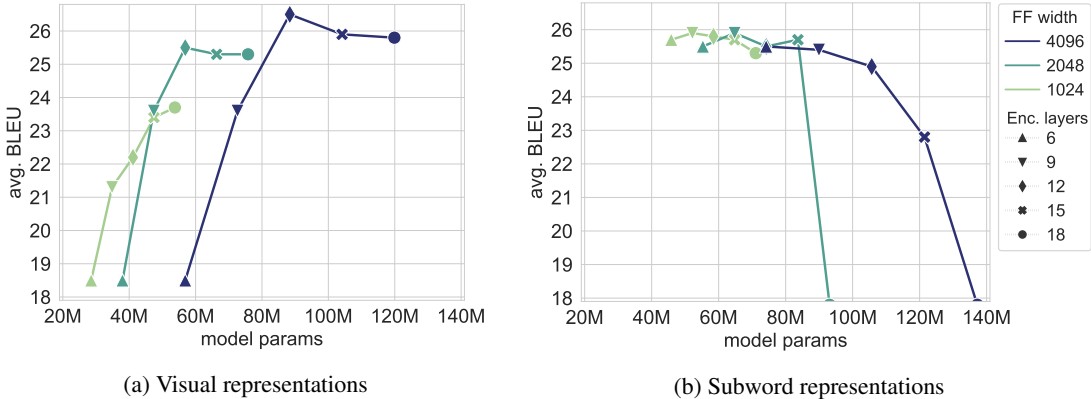

|                    | (a) Visual representations | (b) Subword representations |
|---|---|---|

Figure 3: Performance across different model capacities, varying encoder depth and/or width (TED-7).

4 heads. We train our models with the Adam optimizer (Kingma and Ba, 2015) with linear warm-up, learning rate 5e-4, dropout of 0.1, and label smoothing 0.2. We train with temperature sampling $T = 1.5$ in language-imbalanced settings (Arivazhagan et al., 2019; Shaham et al., 2023). We use batches of 160k tokens, and train until performance on a held-out validation set fails to improve for ten validations. Trained models and scripts to replicate them will be released upon publication.[8]

Reparameterizing model capacity with deeper encoders and shallower decoders has been shown to be beneficial particularly for large multilingual vocabularies and/or smaller granularity inputs such as characters (Cherry et al., 2018; Kasai et al., 2021; Kong et al., 2021; Xu et al., 2021; Berard et al., 2021). Replacing the source embedding matrix with visual representations frees parameters which may be re-allocated elsewhere within the model. As we expand language coverage with pixel-based representations, it is not clear a priori whether and where additional capacity may be needed to scale performance compared to models for individual languages or traditional text models. We experiment with different ways to add and allocate model capacity with both pixel and text inputs, with results presented in § 3.1.

## 3 Multilingual translation with pixels

We experiment with two datasets to investigate the performance and properties of multilingual pixel representations for machine translation. We perform initial experiments with the balanced 7 language pair multi-target TED data (Duh, 2018) used by Salesky et al. (2021), which we will refer to as TED-7, to compare performance to prior work with

pixel representations and explore any necessary architectural changes in the multilingual setting. We then scale up using the larger 59 language pair TED talk corpus from Qi et al. (2018), or TED-59. In all cases, our models are many-to-one multilingual translation models with English as the target. We list the languages in each corpus with the number of training examples in App. A. Results for all datasets are shown in Table 1.

### 3.1 Model capacity: wider or deeper?

Increasing language coverage often requires increased model capacity. We find that the small base architecture from Salesky et al. (2021) is unstable and may not converge when trained multilingually without additional capacity or batch size. For TED-7, multilingual source embeddings account for 33% of the total parameters of the best subword model.[9] Without a source embedding matrix, despite the additional convolutional block, a pixel model with the same Transformer architecture as a subword model would be ~17M parameters (or 38%) smaller, as seen in Figure 3, and may thus require different parameterization and result in different scaling behavior.

We investigate both the impact of reparameterizing the baseline model, as well as increasing capacity through greater encoder depth and/or width. We first find that shifting from an equal depth encoder-decoder model to a deep encoder and shallow decoder with the same number of parameters provides consistent improvements; for example, moving from $6-6$ to $9-3$ improves performance on the TED-7 dataset from 18.5 to 21.3 BLEU (+2.8). We maintain a shallow 3 layer decoder while varying the encoder through the remainder of this section.

---

[8]https://github.com/esalesky/visrep/tree/multi

[9]TED-7 uses a source subword vocabulary of 35k; this disparity would continue to increase with larger vocabulary sizes.

| Source reps. | TED-7 | | | TED-59 | | |
| --- | --- | --- | --- | --- | --- | --- |
| | BLEU | chrF | COMET | BLEU | chrF | COMET |
| BPE | 25.7 | 48.8 | 77.3 | 23.8 | 45.5 | 73.3 |
| PIXEL | 26.2 | 49.5 | 77.9 | 28.4 | 50.1 | 77.2 |

Table 1: Model performance across two datasets on test. Models chosen by perplexity on held-out validation sets. Metric scores are averaged across all languages in the dataset; App. E shows results for individual language pairs.

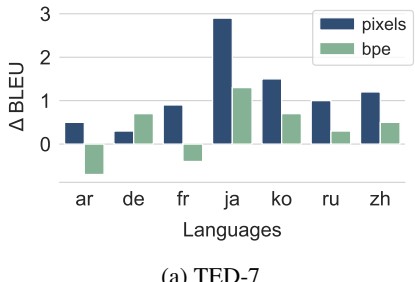

(a) TED-7

Figure 4: Improvement with multilingual models over models for each lang. pair.

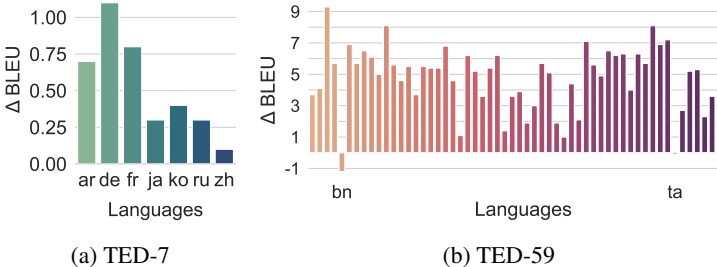

(a) TED-7

(b) TED-59

Figure 5: Improvement per language with pixel representations compared to subwords on multilingual TED-7 and TED-59 datasets.

With an equal number of model parameters, increasing depth is more impactful than width, as seen in Figure 3a. Increasing width provides consistent improvements at all model sizes, while more significantly increasing overall parameters. With 12−3 layers and 2048 FF width, a pixel-based model has an equivalent number of parameters to the best subword model (∼55M) while able to continue improving with scale. The best pixel models for this dataset use 12−3 layers and FF width 4096. Continuing to increase depth and overall size has diminishing returns. Pixel models also appear more robust to overparameterization where text models degrade more quickly, as seen in Figure 3b.

Is the optimal parameterization determined by the granularity of pixel inputs, the amount of training data, or the multilinguality of the task? To see, we reparameterize the models for individual language pairs from Salesky et al. (2021) at both the small and large data sizes (shown in App. F). We find that performance would have decreased in both cases, suggesting this is more likely due to the multilingual task, not the amount of data or pixel representations inherently.

For the larger TED-59 dataset (1.2M→5.1M), we use the same architecture as for TED-7. Exact model configurations for each dataset and representation scheme are listed together in App. B.

### 3.2 Language coverage and imbalanced data

Including additional languages can sometimes interfere with rather than improve performance ('the curse of multilinguality' (Conneau et al., 2020)). When we compare our multilingual models to individual models for the same language pairs with TED-7, we see that *all* languages improve through multilingual training with pixel representations, while this is not the case for subword-based models, where two language pairs degrade (Figure 4). Improvements are greatest for those language pairs (*ja*, *ko*, *zh*) where individual models performed worse than BPE in Salesky et al. (2021). Improvements could be due to boosts from languages with similar scripts (*zh* and *ja*, or *fr* and *de*) or simply an increase in total training data: we investigate this in § 4.1 for TED-59 where we have more languages to study. Notably, improvements come without interference for pixel models here. Comparing multilingual pixel and BPE models, we see small but consistent improvements on TED-7 (Figure 5).

The TED-7 setting has relatively balanced data across all languages and scripts and at least 150k examples per pair, which is a reasonable baseline but unrealistic in the context of typical multilingual translation settings. We turn to the TED-59 dataset for increased language coverage with imbalanced training data and script representation for a more realistic setting to see if our improvements hold or interference emerges. Here we see *larger* improvements of up to 9 BLEU compared to BPE for most language pairs, and some degradation for 2 pairs whose scripts have only ∼5k training examples across all languages, highlighted in Figure 5.

Given the large and imbalanced nature of this

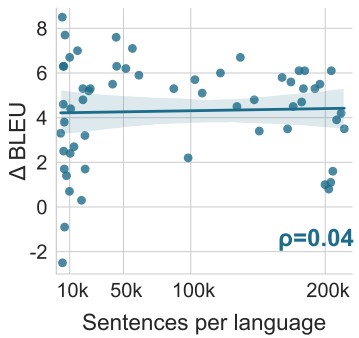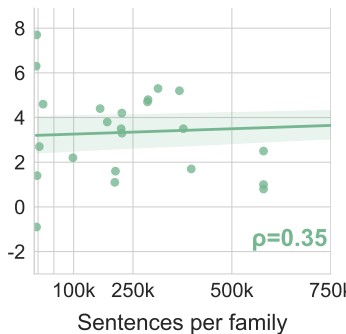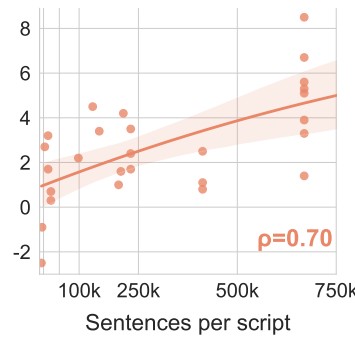

Figure 6: Performance improvements with pixel representations are most strongly correlated with the total amount of data for a language's script compared to language or language family. Data size per language is listed in App. A.

| Src | Script | # Sents | PIXEL | BPE | Aharoni. | Δ |
|-----|--------|---------|-------|-----|----------|------|
| az | Latin | 5946 | 16.6 | 12.5 | 11.2 | +5.4 |
| be | Cyrillic | 4509 | 28.5 | 19.2 | 18.3 | +10.2 |
| gl | Latin | 10017 | 36.5 | 29.7 | 28.6 | +7.9 |
| sk | Latin | 61470 | 33.7 | 27.4 | 26.8 | +6.9 |
| **[ LR ]** | | *avg:* | 28.8 | 22.2 | 21.2 | +7.6 |
| ar | Arabic | 214111 | 29.8 | 26.1 | 25.9 | +3.9 |
| de | Latin | 167888 | 36.1 | 30.0 | 28.9 | +7.2 |
| he | Hebrew | 211819 | 35.3 | 30.7 | 30.2 | +5.1 |
| it | Latin | 204503 | 38.5 | 32.3 | 32.4 | +6.1 |
| **[ HR ]** | | *avg:* | 34.9 | 29.8 | 29.4 | +5.6 |

Table 2: Results for 4 high-resource (HR) and low-resource (LR) language pairs used in previous work.

dataset, previous work has commonly reported non-aggregated performance for a subset of language pairs only (4 low-resource and 4 high-resource) with varied scripts and degrees of relatedness. Compared to the best previous results on those pairs (Aharoni et al., 2019), our subword baselines improve slightly: +1 BLEU on the LR pairs and +0.4 on the HR. With pixel representations, our models improve significantly, +7.6 on the LR pairs and +5.6 on the HR pairs, as shown in Table 2. Low-resource languages with well-represented scripts shared with other languages show larger improvement than the overall mean of +4.6; most dramatically, Belarusian (be) improves by >50% or 10.2 BLEU despite having only 4509 training instances through positive transfer from the >600k sentence pairs in Cyrillic in TED-59 (discussed further in § 4.1). Jin and Xiong (2022) presented the strongest previous performance on the full TED-59 dataset with a many-to-many multilingual model[10] with language-aware multi-head attention, with

---

[10]Their many-to-many model is trained on 2× as many sentences as the models presented here by reversing the dataset.

25.3 average BLEU: our many-to-one pixel model improves on this by +3.1 BLEU. They do not report results per language for further comparison.

## 4 Properties of multilingual pixel models

### 4.1 Positive transfer across languages

We look at the relationship between data representation for each source language, family, script and performance to find the greatest contributors to improvements with pixel representations on TED-59. The amount of data for a given pair is only weakly related to performance for both pixel and subword representations ($\rho \leq 0.3, p < 0.05$), while language family and script representation is moderately correlated ($\rho = 0.5 - 0.6, p \ll 0.001$) suggesting some positive transfer across languages and scripts for both approaches. However, looking at each factor's relationship to performance *improvement* rather than raw scores better reflects those responsible for the difference. As shown in Figure 6, the amount of data for a given script is strongly correlated with ΔBLEU, ($\rho = 0.70, p \ll 0.001$), while family is moderately correlated (0.35) and data for individual language pairs has no clear relationship. We conclude that pixels enable more effective cross-lingual transfer between languages with the same script, and to a lesser degree family, than joint subword vocabularies. We hypothesize that we would see similar improvements for Bengali and Tamil with at least 10k examples for their scripts.

### 4.1.1 Clustering by language and script

To better understand how pixel representations pattern by language and script we compare our model subword embeddings and our pixel representations. Using the validation set as input, we compute sentence-level vectors by mean-pooling over token embeddings for each sentence for the

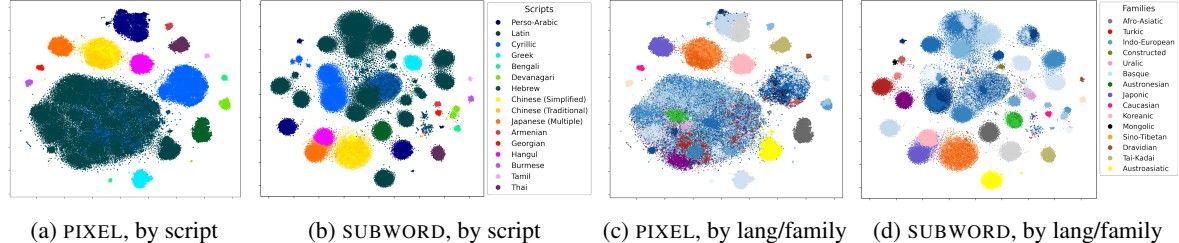

| (a) PIXEL, by script | (b) SUBWORD, by script | (c) PIXEL, by lang/family | (d) SUBWORD, by lang/family |

Figure 7: Clustering shows more representational similarity within scripts and across languages with pixel representations than with disjoint subword embeddings in the TED-59 dataset. Individual languages from the same family are shown with different shades of the same color in items (c) and (d).

subword model, or over the linearly projected vectors of the same dimension from the convolutional block in the pixel model. We visualize these representations using t-SNE clustering (van der Maaten and Hinton, 2008), in Figure 7 for TED-59 and in App. H for the smaller TED-7.

Pixel representations cluster neatly by script (7a), reflecting the strong ability to share information between languages of the same script discussed in § 4.1. Subword embeddings do not cluster as strongly by script despite shared subwords, with many separate clusters for e.g. Latin script languages (7b). We observe that subword embeddings cluster more tightly by language and family (7d), with less representational overlap between languages than we see with pixels (7c). However, the visual model still reflects some similarities within families both within and across scripts. For example, in the large Latin-script cluster in 7c, all Uralic languages appear within close proximity of each other, as do Austronesian, and some overlap exists between Cyrillic and Latin representations in 7a, which likely reflects Slavic family similarities rather than visually similar characters given sentence-level vectors.

## 4.2 Complete parameter sharing

With traditional model vocabularies, parameters are not shared between embeddings; only 3% of embeddings are updated per batch on average[11] for TED-59 without redistribution techniques such as label smoothing. On the other hand, 100% of the pixel model representation block parameters are updated every batch due to parameter sharing at the pixel level. Pixel representations have direct access to token sub-components, whereas subwords do not, leading to more similar representations for

---

[11]Heavily dependent on language coverage, sampling, vocabulary, and batch size. This number reflects a 64k source vocabulary and large batch size of 160k tokens.

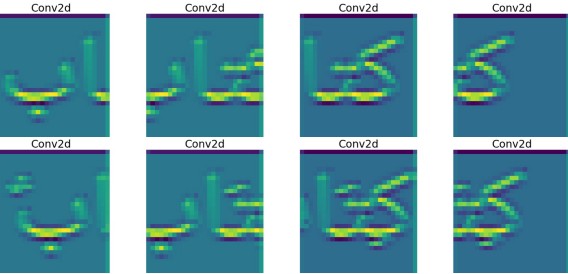

Figure 8: Pixel representations result in similar representations for partial lexical matches due to visual similarity and parameter sharing at the pixel level.

words e.g. with and without diacritics—with the TED-59 subword vocabulary, the Arabic forms كتاب and كِتَاب for "book" have disjoint subword decompositions and so do not share embeddings, whereas the pixel representations are highly similar; as visualized in Figure 8, the convolutional layer feature activations remain highly similar despite the inserted diacritics. If a pixel-based model observes partial lexical matches such as "ktb" and "kitab" in training, parameters for both will be updated by backpropagation to the shared pixel values; we hypothesize that this contributes to the increased transfer across languages with the same script and performance improvements. Future work may investigate whether this property leads to more compositional representations.

## 4.3 Reduced frequency-based representation degeneration

Previous work has shown that embeddings can suffer from a frequency-based representation degeneration problem, where infrequent and unseen words cluster together in embedding space due to limited parameter updates during training (Gao et al., 2019). However, as pixel models share parameters at the pixel level, all representations are updated to some degree each batch regardless of subword-

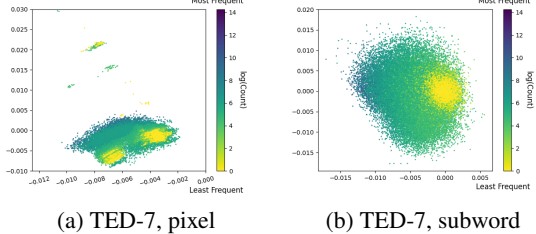

(a) TED-7, pixel        (b) TED-7, subword

Figure 9: SVD plots of source representations show traditional embeddings cluster infrequent subwords together more tightly than pixels.

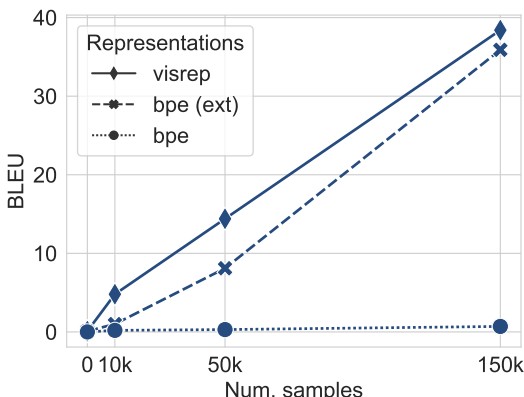

Figure 10: Data-efficiency in cross-lingual transfer. Models with pixel-based representations adapt more efficiently and effectively to new scripts than with traditional text representations (shown here: Hebrew).

| | Script seen? | Unigram Coverage | Bigram Coverage | Trigram Coverage | Gain BPE ext | Gain BPE |
|---|---|---|---|---|---|---|
| ro | ✓ | 96% | 91% | 84% | +11% | +11% |
| pl | ✓ | 95% | 88% | 73% | +13% | +14% |
| fa | ✓ | 99% | 79% | 66% | +18% | +20% |
| vi | ✓ | 86% | 66% | 41% | +22% | +21% |
| he | ✗ | 23% | 5% | 1% | +30% | +4700% |

Table 3: Script coverage in pretraining measured at the level of character $n$-grams. Improvements with pixel representations are averaged across all resource settings.

level text frequency. Therefore, the low-frequency degradation effect should reduce in pixel models and rare words may not cluster as strongly.

We examine this phenomenon by comparing the source embeddings from the subword model against representations from the pixel model on TED-7. We obtain a comparable set of representations from the pixel model by rendering each subword in the TED-7 source vocabulary and mean-pooling the output of the convolutional block for all resulting resulting visual token(s).

We plot these embeddings using 2-D singular value decomposition, and color each point according to the log-frequency of its corresponding subword in Figure 9. We plot visual embeddings, excluding 1% of outliers for improved readability (and include the full plot in App. I). We see that in the text model, there is both a clear frequency bias and and a cluster of low-frequency embeddings. In the pixel model, though we see some frequency bias among embeddings, the distribution of low-frequency embeddings is improved.

## 5 Data-efficient cross-lingual transfer

It has been shown that using pretrained multilingual models for cross-lingual transfer can provide significant performance improvements, particularly when the target language is under-resourced. However, adapting models to unseen scripts with no lexical coverage in the original model typically requires techniques such as expanding the embedding matrix to include new vocabulary (Wang et al., 2019b) or language-specific adapters (Houlsby et al., 2019; Pfeiffer et al., 2020). In contrast, models with pixel representations can be finetuned directly on new languages and scripts without requiring any architectural changes (Rust et al., 2023). We hypothesize that the model properties discussed in § 4 will not only allow transfer without model ex-

tensions, but enable transfer *more data-efficiently*, requiring fewer examples to achieve good performance.

To evaluate the data-efficiency of cross-lingual transfer, we adapt our multilingual models to language pairs with five new source languages, each with different degrees of script coverage to those observed in pretraining as quantified in Table 3: Romanian, Polish, Farsi, Vietnamese, and Hebrew. We randomly sample 10k, 50k, and 150k (~all) sentences from the multi-target TED dataset used for TED-7 for each new language pair and fine-tune our TED-7 models on the training data for each pair individually for up to 30 epochs, with early stopping if there are no improvements on the held-out validation sets for 5 epochs. We use the TED-7 models because they do not cover these languages in pretraining; we note that the overall performance on the original task is similar for pixel and subword models. In addition to the pixel and subword models, we also compare subword models with vocabulary expansion, where the source embedding matrix is extended to include BPE inventories of size 5k trained for each new language,

for which embeddings are randomly initialized.

Whether model vocabularies cover a particular script is typically described as binary, but even with observed scripts new languages introduce unseen character sequences and diacritics which will not be appropriately represented. We observe that for Unicode-based models, transfer capability is strongly reflected in lexical coverage; vocabulary expansion improves performance slightly for languages with higher $n$-gram coverage, and significantly for Hebrew with minimal coverage, particularly with more data to train new language-specific embeddings, as seen in Figure 10. However, pixel representations enable models to perform better still than vocabulary expansion, particularly with less data. We believe this is because with complete parameter sharing across all scripts, all parameters for new languages are more strongly initialized. This direction may lead to more data-efficient cross-lingual transfer, particularly for under-resourced languages and tasks.

## 6 Related Work

Previous work has shown allocating additional encoder capacity to be beneficial for smaller granularity inputs, both for characters and bytes (Cherry et al., 2018; Xue et al., 2022b) and other modalities (He et al., 2021; Zhang et al., 2017). Deep encoders and shallow decoders have been used to improve model efficiency and latency with subword inputs (Kim et al., 2019; Kasai et al., 2021; Kong et al., 2021), and deeper and narrower encoders have been shown to scale more effectively (Tay et al., 2022; Xue et al., 2022a).

Significant prior work has been devoted to broader and more effective language coverage, through full Unicode character coverage and downsampling (Clark et al., 2022), clustered vocabularies for efficient modeling of large vocabularies (Chung et al., 2020; Liang et al., 2023), byte-level modeling (Gillick et al., 2016; Xue et al., 2022b), bytes in conjunction with BPE to combat data sparsity and memory issues (BBPE: Radford et al., 2019; Wang et al., 2019a) or byte-fallback (Xue et al., 2022b). Mapping characters to a smaller set of common representations across scripts through transliteration (Amrhein and Sennrich, 2020; Purkayastha et al., 2023) or grapheme-to-phoneme systems (Sun et al., 2022; Gheini and May, 2019) have also been shown beneficial for multilingual and cross-lingual transfer for re-

lated languages across scripts, though they may also introduce collisions which can negatively affect performance. Post-hoc vocabulary expansion (Wang et al., 2019b; Moon and Okazaki, 2020) or language adapters (Houlsby et al., 2019; Pfeiffer et al., 2020) to increase vocabulary coverage have also been shown to be very effective. Recently, pixel representations have been proposed as a vocabulary-free alternative (Salesky et al., 2021; Rust et al., 2023), though not trained yet multilingually. We refer readers to the BigScience survey for greater discussion (Mielke et al., 2021).

## 7 Conclusions

We introduce and demonstrate how to effectively train multilingual pixel representations for machine translation. We experiment with two different data scales with a variety of language and script coverage, demonstrating improved performance compared to the traditional subword approach. We analyze various properties of pixel representations to better understand where they may provide potential benefits and the impact of different scripts and data representation. We observe that these properties not only enable cross-lingual transfer to unseen scripts, but make pixel representations more data-efficient than alternatives such as vocabulary expansion. We hope this work contributes to more extensible multilingual models for all languages and scripts.

## 8 Limitations

Our multilingual experiments are only many-to-one thus far, and apply visual representations to the source languages only. Whether the dynamics would change with multiple target languages is not yet known. Though we do experiment with multiple resource scales up to ~5M sentences our settings remain limited in scale and domain compared to large-scale industry models and it remains to be seen how this approach would fare in other settings. At very low-resource settings with fewer than 10k examples for a given script, our approach may perform worse than traditional subword embeddings. We observe that pixel models are in some settings slower to converge than subword equivalents, which we cautiously attribute to sub-optimal hyperparameters. Though the compute resources required for training models are similar to traditional text representations, significantly more disk space is required to save rendered text compared to

raw text, which may be necessary if pre-computing batches without rendering on-the-fly and may limit efficiency in larger-scale settings. Scalability to longer text has not yet been investigated.

## 9 Ethics Statement

The aim of this work is to reduce the vocabulary bottleneck which disproportionately affects low-resource languages as they are less likely to be appropriately represented in traditional discrete multilingual model vocabularies. Alternatives such as byte-level tokenization potentially increase rather than decrease the disparity between scripts, as a single character may be represented as up to 12 bytes in e.g. Telugu, whereas Latin scripts are typically 1:1 characters:bytes (Ahia et al., 2023). We show the sequence lengths resulting from byte, character, BPE, and pixel 'tokenization' on TED-59 in Figure 11, App. D; of the alternatives to BPE tokenization, pixel representations result in the most similar sequence lengths and lowest variance across languages and scripts.

In application settings, substituting visually similar characters such as '0' for 'O' can be used to circumvent lexical filtering as used for e.g. spam filtering, hate speech detection, or censorship. Pixel representations may make these substitutions less effective which may be beneficial or harmful depending on the setting.

## Acknowledgments

We are grateful to Team PIXEL (Phillip Rust, Jonas F. Lotz, Emanuele Bugliarello, Miryam de Lhoneux, Desmond Elliott), Carlos Aguirre, Antonis Anastasopoulos, and Chenlei Sei for helpful discussions. Elizabeth Salesky is supported by the Apple Scholars in AI/ML fellowship.

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

## A  List of Languages by Dataset

We list the source languages in each dataset with the number of training examples and language code. All datasets are many-to-one parallel with English as the target language.

For TED-7 and TED-59, we use the provided train/dev/test splits, and report results on test using model checkpoints chosen based on dev perplexities.

| | **TED-7** | | | | | | | |
|---|---|---|---|---|---|---|---|---|
| ar | Arabic | 175k | ja | Japanese | 155k | zh | Chinese | 170k |
| de | German | 153k | ko | Korean | 166k | | **Total:** | **1.2M** |
| fr | French | 158k | ru | Russian | 181k | | | |

| | **TED-59** | | | | | | | |
|---|---|---|---|---|---|---|---|---|
| ar | Arabic | 214k | he | Hebrew | 212k | pl | Polish | 176k |
| az | Azerbaijani | 6k | hi | Hindi | 19k | pt | Portuguese | 52k |
| be | Belarusian | 5k | hr | Croatian | 122k | pt-br | Br. Portuguese | 185k |
| bg | Bulgarian | 174k | hu | Hungarian | 147k | ro | Romanian | 180k |
| bn | Bengali | 5k | hy | Armenian | 21k | ru | Russian | 208k |
| bs | Bosnian | 6k | id | Indonesian | 87k | sk | Slovak | 61k |
| calv | — | 0k | it | Italian | 205k | sl | Slovenian | 20k |
| cs | Czech | 103k | ja | Japanese | 204k | sq | Albanian | 45k |
| da | Danish | 45k | ka | Georgian | 13k | sr | Serbian | 137k |
| de | German | 168k | kk | Kazakh | 3k | sv | Swedish | 57k |
| el | Greek | 134k | ko | Korean | 206k | ta | Tamil | 6k |
| eo | Esperanto | 7k | ku | Kurdish | 10k | th | Thai | 98k |
| es | Spanish | 196k | lt | Lithuanian | 42k | tr | Turkish | 182k |
| et | Estonian | 11k | mk | Macedonian | 25k | uk | Ukrainian | 108k |
| eu | Basque | 5k | mn | Mongolian | 8k | ur | Urdu | 6k |
| fa | Farsi | 151k | mr | Marathi | 10k | vi | Vietnamese | 172k |
| fi | Finnish | 24k | ms | Malay | 5k | zh | Chinese | 6k |
| fr | French | 192k | my | Burmese | 21k | zh-cn | Chinese, Simplified | 200k |
| fr-ca | Ca. French | 20k | nb | Norwegian Bokmål | 16k | zh-tw | Chinese, Traditional | 203k |
| gl | Galician | 10k | nl | Dutch | 184k | | **Total:** | **5.1M** |

## B  Model details by dataset

Below we report the details of the best performing model for each dataset and source representation.

| Dataset | #Sents | Model | $V_{src}$ | $V_{tgt}$ | Emb. dim. | Enc. layers | Dec. layers | FF width | Attn. heads | #Params |
|---|---|---|---|---|---|---|---|---|---|---|
| TED-7 | 1.2M | PIXEL | ∅ | 10k | 512 | 12 | 3 | 4096 | 4 | 87M |
| TED-7 | 1.2M | BPE | 35k | 10k | 512 | 6 | 6 | 1024 | 4 | 55M |
| TED-59 | 5.1M | PIXEL | ∅ | 10k | 512 | 12 | 3 | 4096 | 4 | 87M |
| TED-59 | 5.1M | BPE | 64k | 10k | 512 | 6 | 6 | 2048 | 8 | 82M |

Table 4: Details of pixel and subword model scale variants.

## C Detailed discussion of rendering parameter choices

Below we discuss our rendering choices in further detail and provide pointers to the experimentation in past work we build from.

**Font:**  Following past work (Salesky et al., 2021; Rust et al., 2023) we use the Noto font family, as it has the widest Unicode coverage within a single font or font family known to us. Previous work has used the non-serif font variant: we find a slight performance decrease of 5% with NotoSerif on TED-59, and accordingly stick to NotoSans.

**Patch size and stride:**  Salesky et al. (2021) extensively tune font size, window size, and stride for single language pair translation experiments, and find that performance may degrade for some language pairs with font size <10pt. For this reason, we use font size 10pt. While that work found slight differences in optimal window size (15-30) and stride (5-20), we found no degradation in multilingual performance with uniform window widths and so use uniform values for simplicity. Rust et al. (2023) used smaller square windows of $16 \times 16$, without any patch overlap (*continuous*), for English pretraining and cross-lingual finetuning for classification tasks. In our multilingual translation experiments TED-59, we find an average 10% performance decrease without any overlap (stride $s$ = width $w$). The maximum height of the characters in TED-59 with font size 10pt is 22px, requiring reduced size or truncation to use window size 16pt. A larger window size of 32 fits all characters but increases the proportion of whitespace pixels, and decreases performance by 7%. With window size 24px we were able to fit all characters and diacritics in this dataset, with best overall performance.

**Convolutional block:**  We find an average reduction in performance of 2.8% (28.4 to 27.6 BLEU, $stdev = 0.6$) without a convolutional block, translating directly from pixel values linearly projected to $512\text{-}dim$ vectors, and so use 1 convolutional block in all experiments reported here.

**Additional rendering strategies:**  Lotz et al. (2023) compare additional rendering strategies to decrease the pixel input space through structured spacing (bigrams, words) or monospace fonts where available, and show improvements on both pretraining and cross-lingual transfer to downstream classification tasks, and multilingual QA. It remains to be seen how these strategies would affect translation.

**Rendering backend:**  In addition to the character-level fallback capabilities mentioned in the main text (§ 2.1), the PangoCairo renderer is also more *efficient* than PyGame, with throughput approaching the Rust-based BERT tokenizer without batch processing, as measured in Rust et al. (2023, App. D).

## D  Variance in sequence lengths across tokenizations

Below we show the sequence lengths resulting from byte, character, and BPE tokenization and pixel representations on TED-59. Of the alternatives to BPE, pixel representations result in the most similar sequence lengths and lowest variance across languages and scripts.

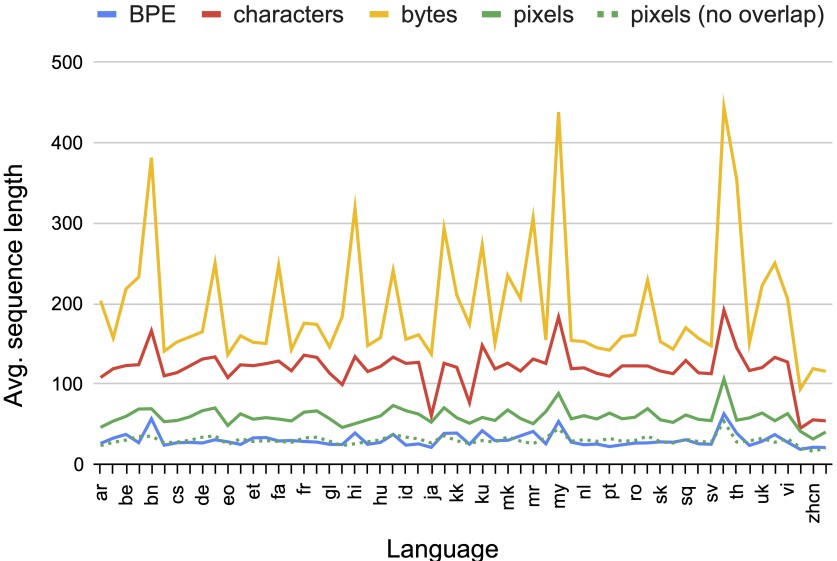

Figure 11: Average sequence length with various tokenization schemes compared on TED-59.

## E  Full results reported by individual language pair

In addition to the aggregated metric scores reported in the main text, below we report results for each individual language pair with three metrics: BLEU, chrF, and COMET.

Results are organized by dataset. TED-7 results are reported in Table 5, and TED-59 in Table 6.

### E.1  Individual language pair results: TED-7

| | **BLEU** | | | | | | |
| | *Mean* | ar | de | fr | ja | ko | ru | zh |
|---|---|---|---|---|---|---|---|---|
| PIXEL | **26.2** | **32.1** | **35.4** | **37.1** | **16.0** | **18.1** | **26.1** | **18.9** |
| BPE | 25.7 | 31.4 | 34.3 | 36.3 | 15.7 | 17.7 | 25.7 | 18.8 |
| CHAR | 24.5 | 30.4 | 32.7 | 34.3 | 15.0 | 16.8 | 24.3 | 17.8 |

| | **chrF** | | | | | | |
| | *Mean* | ar | de | fr | ja | ko | ru | zh |
|---|---|---|---|---|---|---|---|---|
| PIXEL | **49.5** | **54.5** | **57.6** | **58.4** | 40.4 | **42.7** | **49.6** | **43.4** |
| BPE | 48.8 | 53.2 | 56.5 | 57.6 | **40.5** | 42.5 | 48.8 | 42.7 |
| CHAR | 47.5 | 52.4 | 55.0 | 56.0 | 39.2 | 41.1 | 47.6 | 41.4 |

| | **COMET** | | | | | | |
| | *Mean* | ar | de | fr | ja | ko | ru | zh |
|---|---|---|---|---|---|---|---|---|
| PIXEL | **77.9** | **79.6** | **79.1** | **81.6** | 75.2 | **76.6** | **76.5** | **76.6** |
| BPE | 77.3 | 78.8 | 77.9 | 80.9 | **75.3** | 76.4 | 75.7 | 75.8 |
| CHAR | 76.7 | 78.8 | 78.1 | 80.2 | 74.3 | 75.6 | 75.3 | 74.5 |

Table 5: Results on **TED-7** evaluation set reported by individual language pair.

## E.2 Individual language pair results: TED-59

### BLEU

| | Mean | ar | az | be | bg | bn | bs | cs | da | de | el | eo |
|---|---|---|---|---|---|---|---|---|---|---|---|---|
| PIXEL | 28.4 | 29.8 | 16.6 | 28.5 | 39.6 | 12.7 | 38.3 | 30.7 | 44.8 | 36.1 | 38.0 | 32.9 |
| BPE | 23.8 | 26.1 | 12.5 | 19.2 | 33.9 | 13.9 | 31.4 | 25.0 | 38.3 | 30.0 | 33.0 | 24.8 |

| | es | et | eu | fa | fi | fr | fr-ca | gl | he | hi | hr | hu |
|---|---|---|---|---|---|---|---|---|---|---|---|---|
| PIXEL | 41.8 | 23.6 | 21.6 | 26.9 | 22.9 | 40.0 | 35.0 | 36.5 | 35.3 | 21.9 | 37.9 | 26.8 |
| BPE | 36.2 | 19.0 | 16.1 | 23.2 | 17.4 | 34.6 | 29.6 | 29.7 | 30.7 | 20.8 | 31.7 | 21.6 |

| | hy | id | it | ja | ka | kk | ko | ku | lt | mk | mn | mr |
|---|---|---|---|---|---|---|---|---|---|---|---|---|
| PIXEL | 23.7 | 32.1 | 38.5 | 13.2 | 21.7 | 11.8 | 18.1 | 18.4 | 27.2 | 35.3 | 11.0 | 11.8 |
| BPE | 20.1 | 26.7 | 32.3 | 11.8 | 18.1 | 7.9 | 16.2 | 15.4 | 21.5 | 30.2 | 9.1 | 10.8 |

| | ms | my | nb | nl | pl | pt | pt-br | ro | ru | sk | sl | sq |
|---|---|---|---|---|---|---|---|---|---|---|---|---|
| PIXEL | 26.1 | 16.2 | 46.8 | 36.0 | 25.7 | 43.8 | 45.0 | 36.3 | 25.7 | 33.7 | 28.7 | 40.0 |
| BPE | 21.7 | 14.1 | 39.7 | 30.4 | 20.8 | 37.3 | 38.8 | 30.0 | 21.7 | 27.4 | 23.0 | 31.9 |

| | sr | sv | ta | th | tr | uk | ur | vi | zh | zh-cn | zh-tw |
|---|---|---|---|---|---|---|---|---|---|---|---|
| PIXEL | 37.3 | 40.4 | 7.6 | 22.5 | 26.3 | 30.2 | 19.1 | 26.8 | 17.1 | 18.5 | 17.4 |
| BPE | 30.4 | 33.2 | 7.7 | 19.8 | 21.1 | 24.9 | 16.8 | 23.2 | 14.7 | 17.3 | 16.5 |

### chrF

| | Mean | ar | az | be | bg | bn | bs | cs | da | de | el | eo |
|---|---|---|---|---|---|---|---|---|---|---|---|---|
| PIXEL | 50.1 | 51.2 | 38.9 | 50.8 | 60.5 | 32.2 | 59.9 | 53.7 | 64.2 | 57.7 | 57.9 | 53.3 |
| BPE | 45.5 | 47.3 | 33.0 | 40.8 | 55.4 | 34.1 | 53.6 | 48.0 | 58.6 | 51.6 | 53.0 | 45.7 |

| | es | et | eu | fa | fi | fr | fr-ca | gl | he | hi | hr | hu |
|---|---|---|---|---|---|---|---|---|---|---|---|---|
| PIXEL | 62.7 | 46.6 | 44.7 | 49.6 | 44.5 | 60.9 | 56.9 | 58.3 | 55.5 | 41.9 | 59.4 | 49.8 |
| BPE | 57.7 | 41.1 | 38.4 | 45.2 | 38.3 | 55.8 | 51.8 | 52.3 | 51.1 | 41.6 | 53.6 | 44.1 |

| | hy | id | it | ja | ka | kk | ko | ku | lt | mk | mn | mr |
|---|---|---|---|---|---|---|---|---|---|---|---|---|
| PIXEL | 44.8 | 54.2 | 59.6 | 36.9 | 43.1 | 33.6 | 41.4 | 39.7 | 50.2 | 57.8 | 33.3 | 32.0 |
| BPE | 41.1 | 48.4 | 54.1 | 34.9 | 39.5 | 28.4 | 39.3 | 36.2 | 44.1 | 52.1 | 29.4 | 31.5 |

| | ms | my | nb | nl | pl | pt | pt-br | ro | ru | sk | sl | sq |
|---|---|---|---|---|---|---|---|---|---|---|---|---|
| PIXEL | 51.8 | 38.9 | 64.9 | 57.4 | 48.8 | 64.3 | 64.9 | 57.9 | 49.2 | 55.8 | 51.7 | 60.0 |
| BPE | 45.3 | 36.9 | 58.8 | 52.0 | 43.4 | 59.0 | 59.6 | 52.2 | 44.6 | 49.8 | 46.1 | 52.5 |

| | sr | sv | ta | th | tr | uk | ur | vi | zh | zh-cn | zh-tw |
|---|---|---|---|---|---|---|---|---|---|---|---|
| PIXEL | 58.9 | 60.6 | 25.8 | 44.8 | 49.7 | 52.6 | 40.2 | 49.1 | 37.9 | 41.2 | 40.2 |
| BPE | 52.3 | 54.1 | 28.1 | 42.7 | 43.8 | 47.2 | 37.4 | 45.1 | 36.9 | 40.3 | 39.6 |

### COMET

| | Mean | ar | az | be | bg | bn | bs | cs | da | de | el | eo |
|---|---|---|---|---|---|---|---|---|---|---|---|---|
| PIXEL | 77.2 | 77.4 | 74.2 | 76.2 | 82.5 | 62.9 | 84.3 | 80.0 | 83.7 | 81.3 | 81.8 | 78.5 |
| BPE | 73.3 | 73.9 | 66.8 | 67.3 | 78.4 | 67.4 | 78.9 | 74.3 | 79.7 | 76.0 | 78.2 | 72.0 |

| | es | et | eu | fa | fi | fr | fr-ca | gl | he | hi | hr | hu |
|---|---|---|---|---|---|---|---|---|---|---|---|---|
| PIXEL | 83.4 | 74.6 | 74.3 | 77.1 | 76.2 | 82.5 | 82.8 | 81.9 | 79.4 | 70.3 | 82.8 | 77.9 |
| BPE | 79.2 | 69.8 | 69.2 | 73.7 | 70.2 | 78.4 | 77.8 | 76.9 | 75.8 | 71.8 | 77.7 | 72.4 |

| | hy | id | it | ja | ka | kk | ko | ku | lt | mk | mn | mr |
|---|---|---|---|---|---|---|---|---|---|---|---|---|
| PIXEL | 75.5 | 81.0 | 82.3 | 71.9 | 71.1 | 65.1 | 75.4 | 63.7 | 78.4 | 81.4 | 68.3 | 65.3 |
| BPE | 73.7 | 76.0 | 77.4 | 69.7 | 69.7 | 60.5 | 73.5 | 61.5 | 72.3 | 77.0 | 66.1 | 66.2 |

| | ms | my | nb | nl | pl | pt | pt-br | ro | ru | sk | sl | sq |
|---|---|---|---|---|---|---|---|---|---|---|---|---|
| PIXEL | 78.6 | 73.2 | 84.0 | 81.4 | 76.8 | 84.5 | 85.2 | 82.2 | 77.0 | 81.2 | 78.5 | 81.9 |
| BPE | 74.1 | 71.6 | 79.1 | 76.5 | 71.4 | 81.1 | 80.4 | 76.8 | 72.6 | 75.6 | 73.5 | 76.2 |

| | sr | sv | ta | th | tr | uk | ur | vi | zh | zh-cn | zh-tw |
|---|---|---|---|---|---|---|---|---|---|---|---|
| PIXEL | 82.0 | 82.5 | 59.8 | 75.9 | 79.8 | 78.8 | 71.3 | 77.8 | 74.5 | 73.8 | 71.5 |
| BPE | 76.3 | 77.3 | 61.7 | 74.8 | 74.6 | 73.8 | 69.1 | 74.4 | 68.7 | 72.9 | 72.1 |

Table 6: Results on **TED-59** evaluation set reported by individual language pair.

## F Control experiment: Reparameterized models for individual language pairs

Here we reparameterize the TED models for individual language pairs from Salesky et al. (2021) according to our findings in § 3.1, shifting encoder–decoder layer depth from 6−6 to 12−3 and feed-forward width from 1024 to 2048, while maintaining approximately the same number of parameters (55.3M vs. 56.9M). We see that this reparameterization is not optimal for individual language pairs with less data, or the individual de-en language pair with a similar amounts of data to TED-7.

| Model | TED | | | | | | | | WMT |
| | ar | de | fr | ja | ko | ru | zh | *Mean* | de |
| --- | --- | --- | --- | --- | --- | --- | --- | --- | --- |
| Salesky et al. (2021) | 32.1 | 33.6 | 36.7 | 14.4 | 17.0 | 25.4 | 18.3 | *25.4* | 32.9 |
| Reparameterized | 31.0 | 33.5 | 35.6 | 12.6 | 15.4 | 24.1 | 17.0 | *24.2* | 29.4 |

Table 7: Performance differences by reparameterizing models for individual language pairs according to § 3.1.

## G Subword vocabulary constructions

Here we compare different multilingual subword vocabulary constructions for baseline text models on TED-7, with results for bilingual models from Salesky et al. (2021) for comparison.

| Model | ar | de | fr | ja | ko | ru | zh | *Mean* |
| --- | --- | --- | --- | --- | --- | --- | --- | --- |
| Bilingual | 32.1 | 33.6 | 36.7 | 14.4 | 17.0 | 25.4 | 18.3 | *25.4* |
| Characters | 30.4 | 32.7 | 34.3 | 15.0 | 16.8 | 24.3 | 17.8 | *24.5* |
| Joint vocab | 30.6 | 33.8 | 36.3 | 15.6 | 17.4 | 25.1 | 18.6 | *25.3* |
| Separate vocab | 31.4 | 34.3 | 36.3 | 15.7 | 17.7 | 25.7 | 18.8 | *25.7* |

Table 8: Results of different vocabulary constructions for baseline text models on TED-7.

## H Clustering by language and script: TED-7

Below we show the same t-SNE clustering from § 4.1.1 for the smaller multi-way parallel TED-7 validation set. Sentence-level vectors for clustering are creating by mean-pooling token embeddings for both the PIXEL and BPE models. We observe clear clustering by source language in the text model, despite parallel sentences and shared subwords. In the pixel model, we observe multiple clusters per language and script, with greater overlap between languages with shared scripts (French and German).

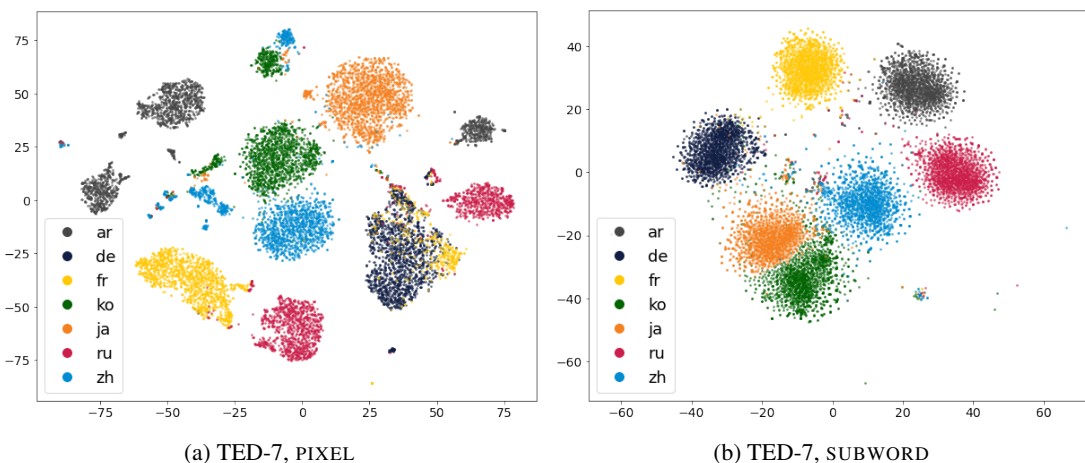

(a) TED-7, PIXEL                    (b) TED-7, SUBWORD

Figure 12: Clustering shows more representational similarity across languages and scripts with pixel representations than with disjoint subword embeddings.

# I  Full SVD plot of TED-7 pixel model embeddings

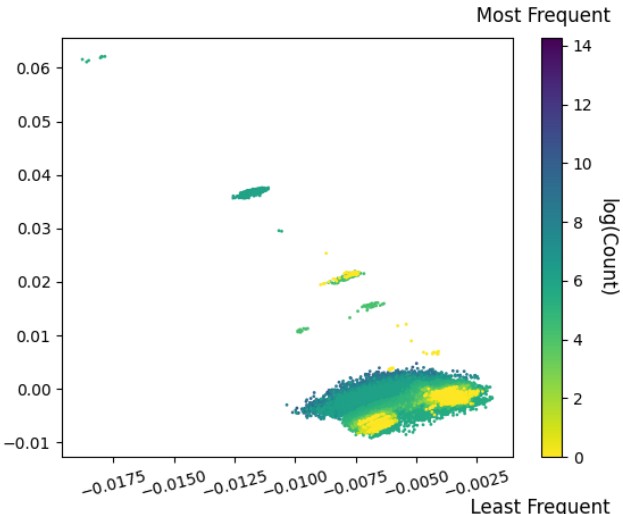

Figure 13: Full SVD visualization of source-side embeddings from the TED-7 pixel model. Only 1% of all embeddings lie above $y = 0.03$, which was excluded from the main text to assist readability.