# OpenReview forum: "Multilingual Pixel Representations for Translation and Effective Cross-lingual Transfer"
_EMNLP/2023/Conference — EMNLP 2023 Main_

### Official Review · Reviewer_mFpr · 2023-07-31

**Typos Grammar Style And Presentation Improvements:** 1. The claim of "up to 9 BLEU" is mis…
**Soundness:** 4

**Excitement:**

4: Strong: This paper deepens the understanding of some phenomenon or lowers the barriers to an existing research direction.

**Paper Topic And Main Contributions:**

The study proposes a method for solving the problem of the limited lexicon of multiligual language models by replacing subword embeddings with image embeddings for the individual script.


**Reasons To Accept:**

That's a well designed study which shows how multilingual models can avoid the lexical bottleneck for lesser-resourced languages.

**Reasons To Reject:**

 1. It is suprising that you haven't experimented with different image conversion parameters, such as different fonts, font sizes and resolutions. Google Noto Sans 10pt at 120 dpi is a very specific setting. The question in Lines 218-220 implies that these variations need to be studied.
 2. Similarly the statement on the source of improvements as in Lines 244-250 should have been studied. Does performance for "ja" improves when you add "zh"?


**Reproducibility:**

4: Could mostly reproduce the results, but there may be some variation because of sample variance or minor variations in their interpretation of the protocol or method.

**Reviewer Confidence:**

4: Quite sure. I tried to check the important points carefully. It's unlikely, though conceivable, that I missed something that should affect my ratings.

---

> ### Author Rebuttal · Authors · 2023-08-25
>
> Thank you for your helpful review! We will clarify these points in the final version.
>
> Reasons to Reject 1:
> Previous work (Salesky et al. 2021, Rust et al. 2023) did this experimentation extensively, so we adapt their best settings. We will clarify why we chose these settings and what experimentation was done in past work in Section 2.1.
>
> Reasons to Reject 2:
> We will clarify the text here. While we do not ablate whether “ja” improves with “zh” only, we do investigate which of the factors mentioned on L244-250 is most critical to positive transfer (script similarities, language similarities, or simply having additional data) for TED-59 where we have more languages to study (Section 4.1), and find that the amount of data for a given script is most strongly correlated with performance improvements.
>
> Presentation 1:
> We will change this to the mean chrF improvement in the abstract and introduction.
> We show chrF but highlight BLEU as it is the only metric reported in previous work on these corpora.
>
> Presentation 2:
> Thank you for this pointer; we have the information necessary to add similar figures to our appendix.
>
> Presentation 3:
> The specific languages in Table 2 were chosen because they are the only languages previous work reported performance for individually in addition to aggregated (average) scores. We note this on L264-267, but will ensure this is clearer in the final version and highlight that this set was originally chosen to include related languages with different scripts as well as unrelated languages with similar scripts.

---

### Official Review · Reviewer_V6mX · 2023-08-04

**Typos Grammar Style And Presentation Improvements:**
**Soundness:** 4

**Ethical Concerns:**

Yes

**Excitement:**

4: Strong: This paper deepens the understanding of some phenomenon or lowers the barriers to an existing research direction.

**Justification For Ethical Concerns:**

please check section 9

**Paper Topic And Main Contributions:**

In this paper, they proposed a novel approach to train NMT model with pixel representations. They evaluated the model on TED dataset with 7 and 49 languages with a variety of languages and script coverage, and improved the perforamnce compared to previous subword-based studies. Further analysis showed that pixel representation helps better cross-lingual transfer to unseen scripts and is more data efficient.


**Questions For The Authors:**

QA: Can you please explain the difference of your approach compared to Rust et al, 2023?

QB: Is it possible to check the approach with better multilingual NMT models?


**Reasons To Accept:**

1. A novel approach to train NMT model with pixel representation
2. Better performance compared to traditional subword-style tokenization
3. The approach also helps better cross-lingual transfer to unseen scripts and is more data efficient

**Reasons To Reject:**

1. The technical difference between their approach and Rust et al, 2023 is not clear
2. Experiment with better multilingual model e.g. NLLB (https://arxiv.org/abs/2207.04672), M2M-100 (https://arxiv.org/abs/2010.11125), and SMaLL100 (https://arxiv.org/abs/2210.11621)?


**Reproducibility:**

5: Could easily reproduce the results.

**Reviewer Confidence:**

4: Quite sure. I tried to check the important points carefully. It's unlikely, though conceivable, that I missed something that should affect my ratings.

---

> ### Author Rebuttal · Authors · 2023-08-25
>
> Thank you for your helpful review! We will clarify these points in the final version.
>
>
> Question A:
> Our rendering approach is similar to the one used in Rust et al 2023. The two key differences are a larger patch size of 24px (in order to fit all characters and diacritics from our broader language set), and overlapping strided patches (we use a stride of 12px, while they do not use overlapping patches) which was previously found beneficial for translation (Salesky et al 2021) which we validated on our settings in initial experimentation.
> Experimentally, the models in Rust et al 2023 were only ever trained on a single language at a time: pretrained on English, and finetuned individually on each target language. The effects of co-training multilingually and whether and why positive transfer or interference may arise with pixel representations were not addressed, nor the data-efficiency of transfer. All experiments in Rust et al 2023 are also classification tasks, not generation.
>
> Question B:
> While we agree that it would be beneficial to compare to larger and stronger models, it would not be possible to directly compare to the pretrained models suggested, as it would require training pixel representations on similarly sized corpora, and their training data is both not fully available and beyond our computational resources.¹
> Adapting pretrained models which were trained with subword embeddings to use pixel representations is a promising future direction but beyond the scope of this work; we will note this in the limitations section of the paper.
> We are running experiments on larger corpora than presented here (50M) which we hope to add.
>
> ¹ SMaLL-100 was finetuned from M2M-100 on a randomly selected 456M sentence subset from CCMatrix which was not released. The original M2M-100 data is 16x larger and expensive to extract from the raw releases. The NLLB training data is not fully public.

---

### Official Review · Reviewer_rKto · 2023-08-05

**Typos Grammar Style And Presentation Improvements:** 121 Please clarify that Table 4 is fo…
**Soundness:** 4

**Excitement:**

4: Strong: This paper deepens the understanding of some phenomenon or lowers the barriers to an existing research direction.

**Paper Topic And Main Contributions:**

Evaluating the use of pixel input representations in many-to-one multilingual machine translation.
While pixel representations are not novel, their interaction with multilingual training has not been studied previously.
Text is rendered as an image using a particular font, and then fed into a convolutional network that transforms it into embedding vectors suitable for a transformer network.

**Questions For The Authors:**

A: 101 On the target side, you decode subword tokens. What is the vocabulary size and construction method for the target subword vocabulary?

**Reasons To Accept:**

The proposed pixel representations improve the crosslingual transfer between languages sharing a script,
as well as enabling transfer between historically related but distinct scripts encoded with different unicode codepoints.
The proposed method is thus especially beneficial for low-resource languages using non-Latin scripts, a disadvantaged group of languages in NLP.

Models using pixel representations are more robust when scaling the model size.

**Reasons To Reject:**

None

**Reproducibility:**

5: Could easily reproduce the results.

**Reviewer Confidence:**

2: Willing to defend my evaluation, but it is fairly likely that I missed some details, didn't understand some central points, or can't be sure about the novelty of the work.

---

> ### Author Rebuttal · Authors · 2023-08-25
>
> Thank you for your helpful review!
>
> Question A:
> We note on L117-121 that we use the same target vocabulary for both pixel and subword input models and list vocabulary sizes in Table 4. Vocabulary construction is SentencePiece unigramLM (L103) and $V_{tgt}$ is 10k for all datasets (Table 4).
> We will add that Table 4 is in Appendix B after it is referenced on L121 and make it clearer that SentencePiece unigramLM is used for all subword vocabularies in the paper.

---

### Meta-Review · Area_Chair_NWHv · 2023-09-18

**Recommendation:** 4

**Metareview:**

This paper presents an empirical study on the effect of pixel-level text representations on multilingual translation models. Building on best practices established in previous work, which was trained one language at a time, this work shows that multilingual pixel representations can improve data efficiency and cross-lingual transfer to unseen scripts. Reviewers are excited about the approach as an alternative to subword models that enables positive transfer for low resource languages, and they have few concerns that were not adequately addressed by the response.

---

### Decision · Program_Chairs · 2023-10-07

**Decision:**

Accept-Main

**Comment:**

This paper presents an empirical study on the effect of pixel-level text representations on multilingual translation models. Building on best practices established in previous work, which was trained one language at a time, this work shows that multilingual pixel representations can improve data efficiency and cross-lingual transfer to unseen scripts. Reviewers are excited about the approach as an alternative to subword models that enables positive transfer for low resource languages, and they have few concerns that were not adequately addressed by the response.